

# The IAGOS $NO_x$ Instrument – Design, Operation and First Results from Deployment aboard Passenger Aircraft

Florian Berkes[1], Norbert Houben[1], Ulrich Bundke[1], Harald Franke[2], Hans-Werner Pätz[1], Franz Rohrer[1], Andreas Wahner[1], and Andreas Petzold[1]

[1] Forschungszentrum Jülich, IEK-8, Jülich, Germany
[2] Enviscope GmbH, Frankfurt, Germany

Correspondence to: Florian Berkes (f.berkes@fz-juelich.de)

**Abstract**

We describe the nitrogen oxides instrument designed for the autonomous operation on board of passenger

aircraft in the framework of the European Research Infrastructure IAGOS (In-service Aircraft for a Global Observing System, www.iagos.org). We demonstrate the performance of the instrument using data from two deployment periods aboard an A340-300 aircraft of Deutsche Lufthansa. The well-established chemiluminescence detection method is used to measure nitrogen monoxide (NO) and nitrogen oxides ($NO_x$). $NO_x$ is measured using a photolytic converter, and nitrogen dioxide ($NO_2$) is determined from the difference

between $NO_x$ and NO. This technique allows to measure at high time resolution (4s) and high precision ($2\sigma$) in the low ppt range (NO of 24 ppt and $NO_x$ of 35 ppt) over different ambient temperature and ambient pressure altitude ranges (from surface pressure down to 190 hPa). The IAGOS $NO_x$ instrument is characterized for (1) calibration stability and total uncertainty (2) humidity and chemical interferences (e.g. ozone, HONO, PAN) and (3) inter-instrumental precision. We demonstrate that the IAGOS $NO_x$ instrument is a robust, fully automated,

and long-term stable instrument suitable for unattended operation on airborne platforms, which provides useful measurements for future air quality studies and emission estimates.





## 1 Introduction

Monitoring of $NO_x$ (= $NO$ + $NO_2$) in the atmosphere is important for estimating the amount of natural and anthropogenic $NO_x$ emissions and for air quality (e.g. formation of ozone and secondary aerosols). The ozone production rate depends strongly on the $NO_x$ mixing ratio, which is most favorable under the conditions

predominating in the upper troposphere (Jeker et al., 2000). Ozone is a strong greenhouse gas and contributes to global radiative forcing (IPCC, 2007; Fahey and Lee, 2016) and to changes of the global dynamics (Fueglistaler, 2014). Close to ground ozone has an impact on human health (Skalska, 2010) and causes ecosystem damage (Ainsworth, 2012), whereas $NO_2$ by itself poses a public health risk as well. Therefore the knowledge of the spatial distribution of $NO_x$ is important to identify the sources, sinks and its partitioning between $NO$ and $NO_2$ in

the atmosphere (Monks et al., 2009).

Most relevant natural sources of $NO_x$ are lightning ($LNO_x$), biomass burning, soil emissions, and anthropogenic sources, such as power generation, road transportation and aviation. The current knowledge of the global distribution of $NO_x$ and its emission estimates is based mostly on specific aircraft missions (Emmons et al.,

1997; Rohrer et al., 1997; Schumann and Huntrieser, 2007; Ziereis et al., 2000; Gressent et al., 2016), on surface monitoring stations (Aerosols, Clouds and Trace gases Research Infrastructure (ACTRIS); www.actris.eu), satellite measurements (Fishman et al., 2008; de Laat et al., 2014; Duncan et al., 2015) and model simulations (Ehhalt et al., 1992; Emmons et al., 1997).

The satellite retrievals provide tropospheric $NO_2$ columns, which are defined as the vertically integrated $NO_2$ number density between the surface and the tropopause. Satellite data users are provided with averaging kernels, which give the relationship between the true vertical profile, and what is actually measured (Eskes and Boersma, 2003). The new experiment TROPOMI on Sentinel-5P provides a global coverage with a spatial resolution of $7 \times 7$ $km^2$. The instrument covers spectral bands in different wavelength which includes bands in the UV spectra

up to SWIR spectra. These bands are selected to measure the most relevant species in the troposphere and to improve cloud correction retrievals (Veefkind et al., 2012). Despite the progress made on modelling aviation's impacts on tropospheric chemistry, there remains a significant spread in model results (Lee et al., 2010). Parameterization of natural NOx emissions by lightning remains still with large uncertainty in global chemical transport models (e. g. Gressent et al., 2016). Brunner et al., (2005) and Prather et al., (2017) concluded that a

better description of emissions, chemistry and sinks of $NO_x$ (and other key species) is needed to improve chemistry in the UTLS region in global chemistry models.

Dedicated aircraft campaigns conducted in the past have made considerable contributions to improve the estimate of the emissions of aviation (Schumann and Huntrieser, 2007; Lee et al., 2010; Wasiuk et al., 2016), the

estimate of $LNO_x$ emissions over different regions, summarized by Gressent et al., (2016), and to increase the knowledge of deep convective lifted pollutants and their burden to ozone chemistry (Huntrieser et al., 2016). However, these and other research aircraft campaigns lack the statistical robustness of comprehensive seasonal and geographical coverage.





Using passenger aircraft as measurement platform, equipped with similar instrumentation can help to link satellite and surface measurements. This is important regarding long-range transport of pollutants and its burden to regional air quality (Petzold et al., 2015). Since 1994, the European Research Infrastructure IAGOS (In-service Aircraft for a Global Observing System, www.iagos.org) provides in-situ observations of essential

climate variables (temperature, water vapor, and ozone, and other species later on), on a global scale from the surface up to 13 km altitude (Petzold et al., 2015). IAGOS builds on the former EU framework projects MOZAIC (Measurement of Ozone and Water Vapour by Airbus In-service Aircraft, Marenco et al., 1998) and CARIBIC (Civil Aircraft for the Regular Investigation of the atmosphere Based on an Instrument Container, Brenninkmeijer et al., 2007). Between 2001 and 2005, total odd nitrogen ($NO_y$ = NO and its atmospheric

oxidation products such as nitrogen dioxide ($NO_2$), nitric acid ($HNO_3$) and peroxyacetyl nitrate (PAN)) was measured on MOZAIC (Volz-Thomas et al., 2005; Pätz et al., 2006) and since 2005 on CARIBIC (Stratmann et al., 2016). Thomas et al. (2015) and Stratmann et al. (2016) presented the geographical distribution and seasonal variation of $NO_y$ at cruise altitude over the different periods. Gressent et al. (2014) showed that the majority of large scale plumes of $NO_y$ are related to long-range transport and only a minor fraction to $LNO_x$ in the upper

troposphere and lowermost stratosphere (UTLS) over the North Atlantic region. Brunner et al. (2001) demonstrate the importance and need of statistical robustness of comprehensive seasonal and geographical coverage of $NO_x$ measurements from one year climatology in the UTLS region, within the framework of NOXAR (measurements of Nitrogen OXides and ozone along Air Routes). However, $NO_2$ was mostly not trustable from these measurements (contamination, instrument failure) at that time and therefore $NO_2$ is based on

calculations of the photochemical state. This accounts also for the CARIBIC platform were $NO_2$ is only available from day time calculation from the photochemical state (Stratmann et al., 2016).

Given its important role in atmospheric chemistry and the resulting needs for global-scale regular measurements, it was decided to develop a $NO_x$ specific instrument for the operation in the framework of IAGOS, which we

describe here. The most common instruments to measure $NO_x$ are based on the chemiluminescence detection (CLD) for indirect measurements of NO (Clough and Thrush, 1967; Ridley et al, 1974, Drummond, et al. 1985; Fahey et al, 1985). CLD instruments have been often coupled with a photolytic or catalytic converter to measure $NO_2$ and $NO_x$ using a xenon lamp, blue-light converter, or catalytic conversion of $NO_2$ using a molybdenum converter (Fehsenfeld et al., 1990; Ryerson et al., 2000; Nakamura et al., 2003; Pollack et al., 2010; Villena et

al., 2012; Reed et al., 2016). $NO_2$ measurements in low $NO_x$ conditions are close to the limit of detection (Yang, 2004), and each instrument might have interferences from nitrogen oxides containing species depending on the used converter (e.g. Reed et al., 2016). Recent instruments are based on optical techniques to measure $NO_2$ with cavity ring down spectroscopy (CRDS, Fuchs et al., 2010; Wagner et al., 2011), cavity attenuated phase shift (CAPS, Kebabian et al., 2008), laser induced fluorescence (LIF, Thornton, 2000) and differential optical

absorption spectroscopy (DOAS, Platt, 2008). However, most of these instruments have a detection limit above 0.1 ppb, or the instrument size and weight is too large to be used for routine aircraft observations (Fuchs et al., 2010; Brent et al., 2015). The technique and design, calibration and quality assurance of the IAGOS Nitrogen Oxides Instrument are presented in Section 2, followed by details about the data processing (Section 3) and the instrument performance (Section 4). First applications of these new measurements aboard an A340-300 aircraft

of Deutsche Lufthansa are presented in Section 5.



**2 The IAGOS NO$_x$ instrument - Package 2b – Measurement system and calibration**

The design of the IAGOS NO$_x$ instrument –Package 2b (P2b) is based on the former MOZAIC NO$_y$ - instrument described by Volz-Thomas et al., (2005) and Pätz et al., (2006) using the chemiluminescence detection (CLD) method for NO with a photolytic converter to convert NO$_2$ to NO. When using a passenger aircraft as platform

many conflicting needs have to be fulfilled. Thus, the instrument has to be fully automated, small and light weight, with limited power consumption, and fulfill high safety standards (mechanical-, strength-tests, and flammability specifications). Furthermore an easy access, simple installation and long deployment periods have to be guaranteed while it should measure scientifically relevant data with the highest possible temporal resolution, accuracy and reliability over the widely varying conditions of external temperature (-70 to +40°C)

and pressure (190 to 1000 hPa) unattended over serval months.

IAGOS NO$_x$ instrument is installed on an IAGOS-CORE mounting rack, which is located in the avionic bay (Fig. 1). The mounting rack provides all electrical, pneumatic and safety provisions required for operation. For data transfer the instrument is connected via Ethernet to IAGOS Package 1 (P1) which handles the data transfer

for all IAGOS instruments on board (Nédélec et al., 2015). P1 is installed on every IAGOS CORE aircraft and provides measurements of ozone, carbon monoxide, temperature, water vapor, and number of cloud particles (hydrometeors). It also records relevant parameters like position, static pressure, velocity, etc. from the avionics of the aircraft (Petzold et al., 2015).

**2.1 The instrument design**

Figure 2 shows the schematic flow and position of the major components of the IAGOS NO$_x$ instrument. The following sections present a detailed description of the detection method (Section 2.1.1), of the reaction cell and the photomultiplier (PMT) as primary detector hosted in the NOD unit (Section 2.1.2), of the ozone generator (O3G), of the photolytic converter (Section 2.1.3), of the inlet manifold (Section 2.1.5), residence time characterization (Section 2.1.6) and of internal stability checks (Section 2.1.7) (ICC). A description of the

instrument operation is provided in Section 2.1.7. The NO detector sensitivity and the converter efficiency are determined in the laboratory (Section 2.2). Table 1 and Table 2 provide an overview of the instrument specification and the main instrument parameters.

**2.1.1 The chemiluminescence detection method**

The CLD method is a well-established technique to detect NO by reaction with excess ozone. NO$_x$ is measured

by converting NO$_2$ into NO. This converted NO$_x$ is often called NO$_c$ at this stage.

$$NO + O_3 \rightarrow NO_2 + O_2 \tag{R1}$$
$$NO + O_3 \rightarrow NO_2^* + O_2 \tag{R2}$$
$$NO_2^* \rightarrow NO_2 + h\nu \ (\lambda > 600nm) \tag{R3}$$
$$NO_2^* + M \rightarrow NO_2 + M (M = N_2, O_2) \tag{R4}$$


In measure mode (in short, MM) the sample air is mixed with ozone in the reaction cell where NO is oxidized (R1) or (R2). The photons released in R3 are detected by a photomultiplier tube (Hamamatsu R2228P or Electron Tubes enterprises 9828A, depending of the individual instrument) which is operated in photon counting



mode. In the zero mode (in short, ZM), ozone is mixed with the sample air before the pre-chamber (a 30 to 50 cm long 1/8" OD SS tube) in order to oxidize most of the NO before it reaches the reaction cell. The volume and thus, the sample residence time of the pre-chamber is adjusted such that 97 to 99% of the NO is oxidized before the sample air reaches the reaction cell. The photon count rate in zero mode includes the background signal of

the photomultiplier (caused by photons originating from the thermal radiation) and additional interferences from other chemical reactions (Drummond et al., 1985). The count rate is quite stable, except during take-off, due to warming up (or cooling down) of different components in the instrument (e.g. ozone generator, PMT etc.). The mixing ratio (X, X ∈ {NO, NO$_2$}) is determined from difference of the photon count rates measured in measure mode and zero mode, divided by the detector sensitivity ($S_{NOD}$) and the conversion efficiency ($E_{PLC}$) in case of

NO$_2$.

$$[X] = \frac{MM - ZM}{S_{NOD} * E_{PLC}}$$                              (Equation 1)

### 2.1.2 The detector and reaction cell

The chemiluminescence detector (in the NOD unit) is similar to that described by Volz-Thomas et al. (2005). The PMT is cooled by four Peltier elements to temperatures below -10°C at an instrument temperature

($T_{Instrument}$=20°C). The reaction cell is separated from the PMT housing by an one mm thick window and a low-pass red filter. This setup provides thermal insulation and limits the light reaching the PMT to wavelengths below 600 nm. The space between the cell window and the low pass filter, as well as the PMT housing, are purged with a small flow of O$_2$ or synthetic air (0.2 ml/min) to avoid condensation. The reaction cell is operated at a pressure of approximately 10 mbar. We learned from the MOZAIC NOy instrument, that the cell does not

require power consuming temperature control because of the relatively stable temperature in the avionic compartment. The temperature is measured, however, in order to allow for potentially necessary corrections of the sensitivity.

### 2.1.3 O$_3$ generator

The ozone is generated in an oxygen flow (approx. 20 sccm) through a ceramic discharge tube with a coaxial

inner stainless steel electrode of 3mm diameter, which is connected to a HV transformer (18 kV). The ceramic tubes are inserted in an aluminum housing which is connected to ground. A silent discharge is generated in the oxygen flow, which produces 1.5E10$^{19}$ molecules per min of O$_3$. The pressure in the discharge tube is kept constant between 1 and 1.2 bar and is monitored by a pressure transducer. More details are described by Volz-Thomas et al. (2005).

### 2.1.4 Photolytic converter

The photolytic converter (PLC) consists of a UV transparent borosilicate glass tube (25 ml), which is mounted behind the manifold. The tube is illuminated by four UV-light emitting diodes (UV-LED, 395± 5 nm, 250mA, 5 VA each, 20 VA total) to convert NO$_2$ in the sample air into NO (R3). The UV-LED's and the associated power transistors of the LED-power-supply are mounted on individual heat sinks which are cooled by air entering

through the bottom of the housing by means of an external fan. The determination of the converter efficiency and the NO$_2$ photolysis frequency ($J_{PLC}$) of the UV-LEDs are shown in Section 2.2. However, laboratory tests



showed that the air passing the PLC is heated by about 30 degrees over the instrument temperature, if the UV-LEDs are switched on (Figure 3). Possible interferences are discussed in Section 4.

**2.1.5 The inlet line, exhaust line and inlet manifold**

The inlet line consists of a 90 cm long with a diameter of 1/8" OD PFA tube. It starts in the Rosemount housing
outside of the fuselage of the aircraft (Nédélec et al., 2015) and ends at the inlet-manifold of the $NO_x$ instrument. The residence time within the inlet line is about 0.05 s thus, losses due to reaction of NO and $O_3$ to $NO_2$ are negligible. About 10% (150 ml/min) of the total inlet flow is sucked from the manifold into the analytic section of the instrument by means of two membrane pumps (Vacuubrand MD1) operated in parallel. The flow is regulated by a mass flow controller (Bronkhorst, IQF-200-AAD-00-V-S). The excess of the inlet flow is flushed
trough the exhaust line, which starts at the end of the inlet-manifold, provided with an exhaust-manifold to gather all flows (e.g. internal calibrations) which passed through the instrument. Outside the instrument the excess flow is guided through the exhaust line (PTFE-tube of 60 cm length with 6 mm outer diameter) to the outlet port at the fuselage of the aircraft. The manifold is made of stainless steel and contains ports for pressure measurement and for the addition of zero air and calibration gas. The total residence time from the manifold to
the NOD is between 2.5 s at cruise altitude and 12 s at sea level. Thus NO losses by reaction (R1) with ozone in the ambient air need to be accounted for when the LEDs of the photolytic converter are switched off.

**2.1.6 Instrument response characterizing**

The response time of the instrument is important for the correction of NO titration by ambient O3 during sampling and by fast changes of the ambient conditions (e.g. the aircraft cross the tropopause). The response
time of the instrument was characterized in the laboratory repeating 10 injections of 2 s NO pulses of 7.1 ppb into the inlet line at each full minute at 250 hPa inlet pressures (Figure 4). The width (1/e) of the NO peak is 4 seconds which represents a peak broadening of a factor two and the delay is about 3 seconds at an inlet pressure of 250 hPa.

**2.1.7 Internal stability checks**

Inside the instrument, $NO_2$ is continuously produced from a permeation tube (PT, KIN-Tek, EL-SRT2-W-67.12-2002/U) placed inside a stainless steel block, which is continuously purged with a small flow (<12 ml/min) of oxygen (Revision 1) or synthetic air (Revision 2). The stainless steel block is temperature controlled at 40±0.5°C using a Pt100 sensor and PID controller. The $NO_2$ flow enters the inlet manifold and is only used for stability checks of the detector sensitivity. During flight, the calibration gas is normally pumped away through the
exhaust and will not reach the sample flow. Disabling this pump flow the calibration gas will reach the analytic section for stability check of about 5 min duration (Figure 5).

**2.1.8 Instrument operation**

The IAGOS $NO_x$ instrument is designed for autonomous deployment over several months. The software utilizes aircraft signals (currently weight on wheels) to switch between operation mode during flight and standby on
ground. The instrument operates in a strict cyclic way by switching the PLC on or off and by flushing the air into the pre-chamber or directly into the reaction cell. During normal operation the ambient air along the flight track





is measured. In addition to the PMT signal (recorded in 10 Hz), pressures, sample flow and temperatures at different positions are recorded as 1-min averages to monitor the state of the instrument. For in-flight system checks, the manifold is flushed in regular intervals with $NO_x$-free gas or $NO_2$ calibration gas (approx. 10-15 ppb, generated from a permeation tube). On ground, the instrument is in standby and not recording data. The ozone

generator (O3G) is off and the valves to the pump and between manifold and exhaust are closed, which leads to a backward flow of synthetic air from the gas bottles through O3G, NOD and manifold to the inlet, in order to avoid contamination when the aircraft is on ground. The different modes of the instrument are summarized in Table 3 and the cyclic measurements during flight are shown in Figure 5.

**2.2 Calibration**

The detector sensitivity, the conversion efficiency and the photolysis rate coefficient are determined by external calibrations in the laboratory using procedures defined in the standard operating procedure (SOP) for P2b (see http://www.iagos.org/iagos-core-instruments/package2b/) and are described in detail in the following subsections. In principle the instrument is flushed with a known mixture of NO and synthetic air, and $NO_2$

produced by gas-phase titration (GPT). The mixing ratio is calculated from the measured flows of the NO calibration gas, oxygen and zero-air. The titration rate of the external GPT mixture is adjusted to 70-90%. A simplified example of one calibration is shown in Fig. S2. Note that the entire calibration procedure is performed at 250 hPa inlet pressure. Table 4 shows the uncertainties of laboratory calibrations for the deployment phase in 2015 and 2016.

**2.2.1 NO detector sensitivity**

The detector sensitivity ($S_{NOD}$) is calculated using the photon count rates ($CAL_{NO}$) by flushing the instrument with a mixture of known NO mixing ratio (µNO) from the secondary standard ( $NO_{Standard}$), synthetic air (SL) and oxygen ($O_2$). Our NO working gas standard (10 ppm NO mixed in $N_2$ (5.0)) is a secondary standard and is regularly referenced to the primary standard of the World Calibration Center for $NO_x$ at the Forschungszemtrum

Jülich. Up to now, deviations between both standards have been smaller than 1%. The uncertainty of the flow measurements is below 2%.

$$S_{NOD} = \frac{CAL_{NO}}{\mu NO}$$                    (Equation 2)

where

$$\mu NO = NO_{Standard} * \frac{flow_{NO}}{flow_{NO} + flow_{SL} + flow_{O2}}$$                    (Equation 3)

The uncertainty of the detector sensitivity ($\delta S_{NOD}$) from the calibrations is 2% to 3% accounting for the errors of the flow meters and primary NO standard.

**2.2.2 NO$_2$ conversion efficiency and the NO$_2$ photolysis frequency**

The conversion efficiency ($E_{PLC}$) is calculated using the measured NO and $NO_x$ signal during the external GPT

by switching the UV-LEDs in the PLC on and off (Table 3). Note, that the instrument background using $NO_x$-




free gas and the signals from the pre-volume (zero mode) must be subtracted from all signals in the measure mode (see Section 3).

$$E_{PLC} = \frac{CAL_{GPT}\,NO_c - CAL_{GPT}\,NO}{CAL_{NO}\,NO - CAL_{GPT}\,NO}$$  (Equation 4)

Typically, the conversion efficiency is between 75% and 85% depending on the ambient pressure. During a deployment period of six months the total uncertainty of the conversion efficiency is determined with 4%, and agrees well with the difference of $E_{PLC}$ between the pre and post calibrations after each deployment.

The photolysis frequency ($j_{PLC}$) of the UV- LEDs is calculated with the following Eq. 5:

$$j_{PLC} = \frac{-ln(1 - E_{PLC})}{\tau}$$  (Equation 5)

with $\tau$, which is the residence time in the converter. The photolysis frequency of the UV-LEDs was stable at $j_{PLC}$ = 0.55 ($\pm$0.05) s$^{-1}$ during the last pre- and post-calibrations at inlet pressure of 250 hPa. During flight, it is used to calculate at each measured data point the conversion efficiency considering the residence time and the ambient pressure in the converter.

### 2.2.3 Zero-air (NO$_x$-free air)

In the laboratory zero air is either generated using

  a)  dried and purified compressed air using a Parker-Hanny adsorption dryer (dewpoint temperature $T_d$ < -40°C) and an additional active charcoal-filter for NO$_x$, ozone and VOCs

  b)  pure O$_2$ (99.5%) from gas bottles, which is also used for the ozone generator

  c)  synthetic air (Air Liquide)

All three zero air types showed in zero mode no differences within measurement errors to each other. This proofs the finding of Volz-Thomas et al. (2005) using the MOZAIC NO$_y$-instrument. However, the difference between measure mode and zero mode of instrument background signal is not equal to zero and has to be subtracted from the ambient measured signal (see Section 3).

### 2.3 Quality assurance

Within the IAGOS community it was agreed to flag data quality according to the criteria elaborated in the FP7 project IGAS (IAGOS for the GMES Atmospheric Service; http://igas-project.de; Gerbig et al., 2014). One major topic of this project was to develop QA/QC rules, defined in SOPs in collaboration with the IAGOS user community. The flagging criteria are summarized in Table 5. Quality assurance is performed according to the SOP for P2b and is described briefly in the following. Shortly, before and after each deployment period, the entire instrument performance is checked and necessary replacements or services of compounds are performed, based on the expected life time of parts or due to deteriorated performance.

The calibration procedure includes:

  •  determination of the detector sensitivity for NO and the conversion-efficiency for NO$_2$ of the PLC using an external calibration setup with GPT

  •  determination of the instrument background with internal zero-air and external zero-air supply

  •  calibration of pressure sensors, capillaries and flow-controllers





Additionally, the in-situ NO measurements are used as in-flight quality check of the instrument since NO is completely oxidized to $NO_2$ during night time and should be zero (see results in Section 5). Internal $NO_2$ calibrations are used to monitor the NO detector sensitivity during the deployment (see Section 4.1). Regular

instrument inter-comparison with state of the art instruments is performed to determine the uncertainty of the instrument (see Section 4.2), which includes case studies for $NO_2$ containing species and their possible interferences (see Section 4.3).

## 3. Data processing

### 3.1 From raw signal to mixing ratio

The following steps describe briefly how the mixing ratios of NO, $NO_2$ and $NO_x$ are calculated from the different measurement modes signal (PMT count rates) for each flight:

1) Interpolate a time series of the different zero-modes signals ($AA\_NOc_{ZM}$ or $AA\_NO_{ZM}$) separately using a running mean with a window size of 400 seconds. This time frame covers at least 4 $NO_c$ and NO mode cycles respectively with the current setup and determines the baseline. The running mean was

chosen because it performed best at the beginning and the end of the time series compared to other interpolation methods.

2) Subtract the interpolated zero modes signal from the measure modes signal (ambient air, zero air, etc).

3) Subtract the instrumental background signals ($BG\_NO_{MM}$ and $BG\_NOc_{MM}$) from the ambient measurement signals ($AA\_NO_{MM}$ and $AA\_NOc_{MM}$) to avoid artifact signals (Drummond et al., 1985).

4) Calculate ambient NO mixing ratio ($[NO]_{AA}$) by applying Eq. 1 and Eq. 2, where $S_{NOD}(t)$ is the time depending detector sensitivity (determined in the lab before and after installation). $S_{NOD}(t)$ is slightly decreasing with time (see Section 4).

$$[NO]_{AA} = \frac{AA\_NO_{MM}}{S_{NOD}(t)} \qquad \text{(Equation 6)}$$

5) Calculate the ambient $NO_2$ and $NO_x$ mixing ratios using the detector sensitivity $S_{NOD}(t)$, the converter

efficiency $E_{PLC}$ and the median NO mixing ratio (before and after each $NO_x$ measurement) applying Eq.1, Eq.2, Eq. 4, Eq.6.

$$[NO_2]_{AA} = \frac{AA\_NOc_{MM} - AA\_NO_{MM}}{S_{NOD}(t) * E_{PLC}(t)} \qquad \text{(Equation 7)}$$

$$[NO_x]_{AA} = [NO]_{AA} + [NO_2]_{AA} \qquad \text{(Equation 8)}$$

6) Apply the water vapor and ozone correction using Eq. 9, Eq. 10 and Eq. 11 (see below).

7) Use night time NO measurements to correct possible offsets associated with the zero mode. Night time periods are identified using the actual position of the aircraft, time and altitude, by calculating the solar zenith angle. Angles larger 100° are used to flag the data as night time. Day time measurements are flagged using solar zenith angles < 80°. In between the measurements are within the twilight zone, where NO is not fully oxidized by ozone.

8) Flag each data point according to Table 5.

9) The data time resolution is provided at four seconds by calculating the median based on 10 Hz raw data for the individual four second periods to be consistent with the other measured compound time series





within IAGOS. The time resolution corresponds therefore to a horizontal resolution of approximately 1 km at cruise altitude. We used the median of the corresponding time interval to avoid a statistical bias uncertainty (Yang 2004).

### 3.1.1 Water vapor correction

The third body quenching effect of water vapor molecules on the excited $NO_2$ molecules in the reaction chamber leads to a reduced signal depending on the amount of ambient water (Parrish et al., 1990; Ridley et al., 1992). The correction factor has to be applied using Eq. 9:

$[NO_{corr}]_{AA}=[NO]_{AA}*(1+\alpha*[H_2O])$    (Equation 9)

With $[H_2O]$ being the water vapor mixing ratio in parts per thousand. In the laboratory we determined the humidity interference parameter of $\alpha=(2.8\pm0.1)*10^{-3}$, independent if the PLC was switched on or off, which is 35% lower than the value of $\alpha=4.3*10^{-3}$ determined by Ridley et al. (1992). Most of the IAGOS data are obtained at cruise altitude, where $[H_2O]$ is in the range of < 5 to 100 ppm. Under these conditions, the water vapor interference is negligible. Within the planetary boundary layer, especially in the tropics, the $[H_2O]$ can

reach values of several thousand of ppm, leading to an interference of up to 10% (Fig. S4). If a water vapor correction could not be applied (e.g. missing water vapor measurements) than the data within the PBL (lowest 3 km above ground) are flagged as "limited" (Tab. 5).

### 3.1.2 Ozone correction

Within the sample line and the converter, reaction (R1) is still active. Depending on the residence time the

reaction will lead to an enhanced $NO_2/NO$ ratio. The residence time ($\tau$) in the inlet is in the order of about 0.05s, and corrections are negligible here. The residence time of the constant sample mass flow within the PLC is about $\tau$=2.5 to 12 s as function of the ambient pressure. The ozone corrections are applied using the in-situ ozone measurements from Package 1, and the photolysis frequency $J_{PLC}$ of the UV-LEDs (see Eq. 5-7) as described in the SOP for $NO_x$ from ACTRIS (Aerosols, Clouds, and Trace gases Research InfraStructure Network,

www.actris.net).

$$[NO]_0 = [NO]_{AA} * \exp(k_{O3} * \tau)$$    (Equation 10)

$$[NO_2]_0 = \left(\frac{J_{PLC}+k_{O3}}{J_{PLC}}\right) * \frac{[NO_c]_{AA}-[NO]_{AA}*\exp(-J_{PLC}*\tau)}{1-\exp(-(k_{O3}+J_{PLC})*\tau)} - [NO]_0$$    (Equation 11)

Here $[NO]_0$ ($[NO_2]_0$) is the expected mixing ratio at the entrance of the Rosemount and $[NO]_{AA}$ ($[NO_2]_{AA}$) is the calculated concentration using the photon count rate, photolysis frequency of the $NO_2$ converter and NO detector sensitivity (see Section 2.2). The factor $k_{O3}$ (= k*[O3]) is calculated from the reaction constant for R1 (k = 1.4E-12*e$^{-1310/T}$, Atkinson et al., 2004), the in-situ measured ozone mixing ratio ([O3]) measured by the IAGOS P1 instrument, and the ambient pressure. Figure 6 shows the correction factor for NO ($NO_{corr}$ = $[NO]_0/[NO]_{AA}$) and

for $NO_2$ ($NO_{2corr}$ = $[NO_2]_0/[NO_2]_{AA}$). NO increases up to 25% and $NO_2$ vary in the range ±10% both depending on the ambient mixing ratio of ozone, temperature, and pressure. Since the ozone correction is very sensitive to





the ozone mixing ratio, we plan to keep the residence time in the PLC at 3 s, independent from the inlet pressure, using a critical nozzle in the next instrument revision in the future.

### 3.2 Instrument uncertainty

#### 3.2.1 Signal precision and limit of detection

The precision of the instrument is limited by the dark noise of the PMT caused by counting thermal radiation photons. The counting statistic is Poisson distributed. The background signal is subtracted from the ambient signal (See Section 3.1). Therefore, the limit of the detection (LOD) is calculated from the $2\sigma$ statistical precision of the zero-air measurements in the measure mode ($BG_{O2}\_NO_{MM}$) and zero-mode ($BG_{O2}\_NO_{ZM}$) which are integrated over four seconds (t=4 s) following Feigl (1998)

$$LOD_{NO} = \frac{2}{S_{NOD}} \times \left( \sqrt{\frac{BG_{O2}\_NO_{MM}}{t}} + \sqrt{\frac{BG_{O2}\_NO_{ZM}}{t}} \right) \qquad \text{(Equation 12)}$$

$$LOD_{NOc} = \frac{2}{E_{PLC} \times S_{NOD}} \times \left( \sqrt{\frac{BG_{O2}\_NOc_{MM}}{t}} + \sqrt{\frac{BG_{O2}\_NOc_{ZM}}{t}} \right) \qquad \text{(Equation 13)}$$

here the different count rates of the photons are given in counts per seconds ($s^{-1}$), and the unit of the instrument
sensitivity is counts per second per ppt ($s\ ppt^{-1}$). We derive a detection limit of $LOD_{NO}$ = 24 (21) ppt for NO and $LOD_{NO2}$ = 35 (30) ppt for $NO_2$ for 4 s integration time for a sensitivity of 0.9 (1.2) s ppt$^{-1}$. By integrating the data over 1 minute, the detection limit improves to $LOD_{NO}$ = 6 ppt and $LOD_{NO2}$ = 9 ppt.

#### 3.2.2 Total uncertainty

The total uncertainty for each measurement point is calculated by error propagation following from Eq. 1:

$$D_{NO} = \frac{1}{S_{NOD}} * (\delta MM + \delta ZM + \delta offset + (MM - ZM - Offset) * \delta S_{NOD}/S_{NOD}) \qquad \text{(Equation 14)}$$

$$D_{NO2} = \frac{1}{E_{PLC} S_{NOD}} * (\delta MMc + \delta ZMc + \delta offset + (MMc - ZMc - Offset) * (\frac{\delta S_{NOD}}{S_{NOD}} + \frac{\delta E_{PLC}}{E_{plc}})) \quad \text{(Equation 15)}$$

The uncertainty of the count rate in measure mode ($\delta MM$), zero mode ($\delta ZM$) and offset ($\delta offset$) is determined from the baseline noise for NO and $NO_x$ measurements. Statistical precision ($2\sigma$) of an individual 4s data point is calculated by error propagation using Eq. 4 and Eq. 5.The uncertainty of the detector sensitivity
during calibration is 2% to 3% and the uncertainty of the converter efficiency is 4% to 5%. Figure 7 shows the relative uncertainty (ratio of the total uncertainty to its measured value) as function of NO and $NO_2$ in the range of measurements observed during 2015. The relative uncertainty of an individual 4s data point is depending on the ambient mixing ratio, and reaches for NO: 25% at 0.2 ppb and 8% at 1 ppb. For $NO_2$ the relative uncertainty is: 50% and 18%, respectively. Similar uncertainties were calculated for all flights in 2016. The total uncertainty
in the low ppt range is mostly dominated by statistical precision of the signal detector.





## 4. Instrument performance

The quality of the IAGOS NO and $NO_2$ measurements depends on the knowledge of the detector sensitivity during the flight phase, the accuracy and precision of the instrument and possible interferences. These issues are discussed in the following subsections.

### 4.1 Instrument performance drift during deployment

The IAGOS $NO_x$ instrument showed regularly a negative drift of the detector sensitivity during each deployment period of two counts per ppb per day. This sensitivity drift was related to a slow degradation of the surface of the reaction cell during the deployment period. The sensitivity losses were corrected by applying a robust linear fit interpolation of the sensitivity between the pre- and post-deployment calibrations. The robust linear interpolation is confirmed from the internal stability checks of $NO_2$ during the deployment phase (Figure 8) and well documented from the MOZAIC $NO_y$ measurements (Thomas et al., 2015). The internal stability checks of $NO_2$ are not further used for determining the mixing ratios from the raw signal. It should be noted, that final data (L2) are provided after the post-calibration. Therefore the instrument operation period is kept short to several months.

### 4.2 Instrument inter-comparison

The German Weather Service organized an inter-comparison of instruments measuring $NO/NO_2/NO_x$ mixing ratios within the framework of ACTRIS. Here 11 European laboratories participated with 17 different state of the art $NO$-, $NO_2$- and $NO_x$ instruments during a two weeks period in October 2016. Most of the time all instruments agreed well and the results of this workshop will provide detailed cross-sensitivities of each individual instrument compared to the reference CLD instrument of the World Calibration Center (WCC) – $NO_x$. The WCC-$NO_x$ instrument (here after REF) was regularly calibrated during this campaign and is used as reference.

Figure 9 shows correlations of NO and $NO_2$ for the IAGOS NOx and the REF instruments for ambient air measurements during two days of this campaign. The ambient air was distributed by a manifold of about 20 m ring-line, with residence times of approximately 5 to 6 s from the first to the last instrument and corrected for ambient ozone mixing ratio. Mixing ratios of NO were observed in the range of the detection limit and 6 ppb. The correlation coefficient is higher than $R^2 > 0.98$ with a slope close to one and an offset of -18 ppt. $NO_2$ was observed in the range of 0.5 to 10 ppb with $R^2 > 0.94$ the slope is close to one with an offset of -102 ppt. The $NO_2$ correlation is larger scattered than NO which is related to the different cyclic measurement of NO and $NO_2$ by both instruments. Further results (e.g. chemical interferences) will be presented in a separate paper. This and future inter-comparisons will assure the quality of the IAGOS $NO_x$ instrument.

### 4.3 Possible interferences

#### 4.3.1 Photolytic decomposition

It is known that photolytic decomposition of HONO can occur when using a photolytic converter for the detection of $NO_2$ with CLD instruments (e.g. Fehsenfeld et al., 1990). During the ACTRIS $NO_x$ side-by-side inter-comparison the interference of HONO within the IAGOS $NO_x$ instrument has been determined to be about 10% at 11 ppb. In-situ observations of HONO in the UTLS regions are very rare and they report only a few ppt





(Jurkat et al., 2011; Jurkat et al., 2016). Thus, the interferences are mostly below the total uncertainty for NO and $NO_x$. This is also the case for $BrONO_2$ and $NO_3$. Both species can be decomposed within the photolytic converter. The concentrations of both species are too small (< 10 ppt) in the UTLS region thus, we expect no major impact on the $NO_2$ measurements (Avallone et al., 1995; Brown et al., 2007; Carslaw et al., 1997).

**4.3.2 Thermal decomposition of $NO_2$ containing species**

The instrument temperature is measured and varies mostly between 15 to 22°C during flight. Close to the surface the instrument temperature can rise up to 30°C in summer. However, the gas temperature inside the PLC increases when the LEDs are switched on. Laboratory measurements showed that the gas temperature in the converter is in the range of 40°C to 70°C at an instrument temperature of 30°C - 35°C (Figure 3). From these

experiments, we extrapolated a gas temperature inside the converter between 27°C (300 K) and 47°C (320 K) during flight. As a result, thermal decomposition of reservoir species containing $NO_2$ can lead to erroneously enhanced $NO_2$ measurements. Reed et al., 2016 showed that the PAN interference could be up to 8 and 25% using a cooled- and not active cooled photolytic converter respectively. In the laboratory, we found $NO_2$ enhancements by mixing 30% of PAN to the sample flow (at 35°C instrument temperature and pressure level of

250 hPa), which was quantitatively generated from a NO calibration gas by photolysis of acetone (100 ppb) in a flow system (Pätz et al., 2002; Volz-Thomas et al., 2002). The result is in good agreement with theoretical calculations of the life time of PAN at the maximum temperature 340 K (at 250 hPa) in the PLC, which predicts an interference of 27%. However, temperature in the PLC above 320 K are not expected during flight, because instrument and units temperature are much lower than in the laboratory, and then PAN interferences should be

less than 3%. Table 6 provides an overview of possible interference to the $NO_2$ measurements over different temperature ranges of the typical reservoir species containing $NO_2$ (dinitrogen pentoxide ($N_2O_5$), peroxynitric acid ($HO_2NO_2$, only during day time), methyl peroxy nitrate ($CH_3O_2NO_2$), and peroxyacetylnitrat (= PAN, $CH_3CO_3NO_2$) at cruise altitude (250 hPa).

**5. First results of NO, $NO_2$ and $NO_x$ observations during inflight operation**

Nitrogen oxides measurements were obtained from two flight phases onboard the Lufthansa Airbus A340. The compiled flight tracks are shown for both years in Figure 10. The aircraft conducted 262 flights in 2015, mostly on routes across the North Atlantic (Düsseldorf – New York or Chicago). In 2016, 208 flights were performed, whilst most flights were on routes from Germany (Frankfurt am Main) to South America (Bogota or Caracas) and various destinations in East and Southeast Asia. In 2015, data of 62 flights are missing due to instrument

shut down because of malfunctioning of system components. Only ten flights are missing in 2016. In total, about 400 hours of data are available in 2015 and 470 hours of data in 2016. The relative amount of the archived measurements with the respective validation flag for all flights is summarized in Table 7. At this stage, parts of the IAGOS measurements are available only as level one data (preliminary) which explains the large fraction of limited data in 2016. Progression of the data to level two (final) is ongoing. Here, we show the first results as

examples, to demonstrate the performance of the instrument. A detailed analysis will be presented in a separate paper if all data are finalized.





### 5.1 NO and $NO_2$ partitioning in the UTLS region

Figure 11 shows the NO and $NO_2$ probability density functions during all night time flights at cruise altitude (p < 350 hPa). NO is expected to be zero which can be confirmed within a variation of about ±25 ppt, which is equal to the statistical precision of the instrument at 4 s time resolution. The quality of the IAGOS NOx measurement

is not only determined by the instrument precision, but also by the homogeneity and representative of the climatological data set. Therefore, the NO measurements at night time are used as an additional quality check during each flight. Sometimes, a small negative NO offset is found (NO < -10 ppt), which occurs due to subtraction of the zero air from the net signal at very low mixing ratios of NO and $NO_x$. However, the half width of the distribution is larger than the random noise of the detector and therefore the value is assumed to be zero.

The median of $NO_2$ is 138.6 ppt with a width range from zero to serval hundreds of ppt. Comparable night time median $NO_2$ value (141 ppt) was observed for the 2016 in the UTLS region. During day time, NO recovers by the photochemical balance with $NO_2$, which leads to a median distribution for NO of 57 ppt (86 ppt in 2016), and for $NO_2$ of 78 ppt (47 ppt). The sum of day time NO and $NO_2$ in 2015 is only 1% smaller compared to the night time $NO_2$ median, which is equivalent to $NO_x$. Difference of day time NO and $NO_2$ between 2015 and

2016 are related to different flight routes and flight levels (Figure 10).

The $NO_x$ partitioning is now compared to previous observations obtained by NOXAR and by CARIBIC. Brunner et al. (2001) showed median $NO_x$ of around 140 ppt (96 ppt) for summer (autumn) in the UTLS region over the North Atlantic in 1995 and 1996. The authors calculated $NO_x$ with the photochemical balance using only day time observations of NO and ozone. These median $NO_x$ values can be confirmed by splitting the

IAGOS measurements in 2015 into summer (165 ppt) and autumn (84 ppt), where the differences between the $NO_x$ medians are less than 15%. The $NO_x$ values from CARIBIC bases also calculated with the photochemical balance using only day time observations of NO and ozone, and only tropospheric air was considered (Stratmann et al., 2016). In summer the median $NO_x$ partitioning is close to 200 ppt and in autumn 100 ppt, which is approximately 16% larger than the IAGOS measurements.

The median of the IAGOS $NO_x$ measurements agree well with the calculated median mixing ratios of $NO_x$ from NOXAR and CARIBIC. However, previous studies identify an unexplained imbalance between the measured and calculated $NO_2$ in low $NO_x$-regions, which were explained with to interferences by $NO_2$ containing species and the large uncertainty of the calculations (e. g. Crawford et al., 1996; Reed et al., 2016). This leads to the

suggestion that interference by $NO_2$ containing species is small for the IAGOS measurements and will be investigated in further studies.

### 5.2 Discussion of observed features in the UTLS

As a first showcase of what can be gained from the IAGOS $NO_x$ observations, Figure 12 demonstrates a time series of all measured compounds on the flight from Düsseldorf to New York City on August 23, 2015. The

measurements (CO, $O_3$, NO, $NO_2$ etc.) are presented as 2 min median averages to reduce the noise, and the potential vorticity (PV) was calculated using ECMWF (European Centre for Medium-Range Forecast) ERA-Interim (Dee et al., 2011) interpolated along the flight track (Berkes et al., 2017). NO varies around zero pptv during night time as expected, while one distinct strong peak of $NO_2$ is observed at 11.5 km altitude which lasts for more than an hour and is correlated with CO and relative humidity. These large peaks are mostly observed





above the local tropopause, which can be identified by the chemical and dynamical tropopause heights. Ozone mixing ratios vary between 100 and 200 ppb and are mostly larger than 150 ppb, while the ozonopause is often reported at 120 ppb (Thouret et al., 2006; Sprung and Zahn, 2010). The location of the dynamical tropopause varies between 2.5 and 5 PVU which is above the commonly used 2 PVU defined location of the dynamical

tropopause for the mid-latitudes (e.g. Kunz et al., 2011).

We focus now on the first more pronounced peak starting at 23 UTC. The timely coincidence with high CO and $H_2O$ values indicates that this air mass is highly polluted compared to typical mixing ratios at this altitude. The origin of this peak was identified using the FLEXPART model. Here a rapid vertical transport from the surface

by deep convection of a long-range transported biomass burning plume could be identified. The FLEXPART model (version 9.02) was used to identify the region with the largest contribution from the surface using five day backward simulations from the particles dispersion (Stohl et al., 2005). FLEXPART results showed that a surface-based air mass was lifted from the North West US within the last 4 days. Here near surface emissions of NO and $NO_2$ from biomass burning could be identified using fire count maps from satellite images during that

time (Fig. S3). These fire emissions contributed also largely to poor air quality in the mid-US at that time (Creamean et al., 2016; Lindaas et al., 2017). Further analyses are beyond the scope of this paper, whereas this showcase study already indicates the possibilities for air quality studies using the full amount of IAGOS observations.

### 5.3 Vertical profiles

Satellite column observations allow monitoring $NO_2$ on a global scale, but the columns do not provide vertical resolution within the troposphere (although there have been recent cloud-slicing methods giving satellite $NO_2$ profiles on a climatological basis), and that the satellite retrieval depends on assumptions on the vertical distribution of $NO_2$ (Bucsela et al., 2008; Boersma et al., 2011; Veefkind et al., 2012). Laughner et al. (2016) showed that the estimates of $NO_2$ at the surface can be largely uncertain in regards to the daily meteorology, if

the priori profile for $NO_2$ is not well known. So far, only a few methods exist to provide in-situ $NO_2$ profiles, however with some limitations (e.g. Piters et al., 2012). We believe that this assumption can be evaluated with in-situ vertical profiles of $NO_2$ from IAGOS to improve the satellite retrievals, which has been successfully demonstrated for CO (de Laat et al., 2014) and ozone (Zbinden et al., 2013).

In total, more than 400 downward profiles of nitrogen oxides are currently available over several regions in 2015 and 2016 (Figure 10). Figure 13 shows statistical analysis of NO and $NO_2$ only at day time over Düsseldorf airport in summer (JJA) 2015. The vertical averaged was calculated in 50 hPa intervals from 200 to 1000 hPa. Median NO ($NO_2$) values reach up to 200 ppt in the UTLS region (9-12 km), which agrees well to the previous observations over the eastern North Atlantic shown by Ziereis et al., (1999, 2000). The median NO ($NO_2$) values

in the mid-troposphere (5 to 9 km), where no major sources exist, vary between the detection limit and 100 ppt. The largest values of nitrogen oxides are measured near the surface with values up to several ppb. It should be noted that these values represent a high polluted region with a huge amount of vehicle-, industry- and aircraft-emissions. The unique IAGOS $NO_2$ profiles will be used for new satellite mission (TROPOMI, www.tropomi.eu) and model evaluation (e.g. air quality) in further studies.





### 6. Discussion and conclusion

The IAGOS $NO_x$ instrument (P2b) setup provides measurements of nitrogen oxide with a good precision and accuracy, while it is highly limited by safety aircraft considerations and its performance by unattended deployment over several months. We presented the different components and the determination of the uncertainties. The relative uncertainty of an individual 4 s data point is depending on the ambient mixing ratio, and reaches for NO: 25% at 0.2 ppb and 8% at 1 ppb, and for $NO_2$: 50% and 18% respectively.

So far only a few instruments are available, which could be used for unattended aircraft observations over several months, because of the need of a high temporal resolution and a low detection limit and fulfillment of the safety considerations. The IAGOS $NO_x$ instrument has a shorter residence time (at cruise altitude) and much larger conversion efficiency of $NO_2$ to NO than instruments using xenon lamps in the 90's, which improves dramatically the instrument accuracy (Ryerson et al., 2000). The detection limit of the IAGOS $NO_x$ instrument ($LOD_{NO}$=24 ppt and $LOD_{NO2}$= 35 ppt at 4 s, 2σ and 0.9 s $ppt^{-1}$ detector sensitivity) is in the range of state of the art instruments used in research aircrafts (e.g. CLD technique: $LOD_{NO}$=10 to 50 ppt and $LOD_{NOx}$= 30 to 80 ppt at 1s (Pollack et al., 2012), CRDS technique (1s, 2σ): $LOD_{NO}$ = 140 ppt, $LOD_{NO2}$=90 ppt (Wagner et al., 2011)). Further instrument inter-comparison in the laboratory and in the field will be performed to reduce the overall uncertainty of the measurements, which are largely limited by the space inside the instrument, the certification and the installation process.

A major advantage of the IAGOS $NO_x$ instrument is the provision of NO and $NO_2$ in-situ measurements at global scale with statistical robustness of comprehensive seasonal and geographical coverage in the UTLS region, and the measurements of vertical profiles from cruise altitude down to the surface over different continents. The statistical analysis over the North Atlantic region shows lower median mixing ratios of NO and $NO_2$ in the UTLS compared to previous projects, where $NO_2$ was determined with the photochemical balance, which is an indication that the possible interferences might be small, if the amount of NOx didn't change over the recent years.

Possible interferences for NO from HONO could be estimated to 10%. The water vapor quenching effect on the NO signal was determined in the laboratory and is applied to the in-situ measurements if water vapor measurements are available. Note, most of the time the aircraft samples in very dry air, where the correction is negligible. However, close to the surface the water vapor correction factor increases up to 10% at 30000 ppm. We apply to the measurements pressure and temperature depended ozone corrections, which has large effects on NO (up to 25%). Thermal decomposition of $NO_2$ containing species might play a major source of uncertainty to the observed $NO_2$ mixing ratios. This includes also the blue light converter, where we aim to reduce the temperature dependency, while it is switched on and off and within the next instrument revision.

The global distribution of $NO_x$ in the UTLS region in combination with transport model calculation allows calculating impact ratios of anthropogenic compared to natural emissions and the concurrency of large scale plumes. This will lead to a better understanding of the ozone chemistry in the very sensitive climate region UTLS. Vertical profiles of $NO_2$ show the expected C-shape profile and the near surface data can be used to



monitor air quality in the vicinity of airports. Further, the day to day variations can be provided to improve satellite a priori profiles in the future (TROPOMI, http://www.tropomi.eu/).

The current setup of the IAGOS NO$_x$ instrument provided more than 800 h of observations and 400 profiles
using only one passenger aircraft as platform within two years (each 6 months). In the near future the number of aircraft will increase, leading to a larger statistical robustness of comprehensive seasonal and geographical coverage of in-situ NO and NO$_2$ measurements.

**Acknowledgements**

The authors are grateful to Deutsche Lufthansa AG for providing special certification and for the transport free
of charge of the instrumentation on one A340-300, and in particular to Gerd Saueressig and to Markus Huf. We also thank enviscope GmbH, in particular Ralf Stosius and Gomolzig for their excellent and continuous support within the IAGOS project. We gratefully acknowledge the continuous support by Andreas Volz-Thomas during the final development of the instrument, and during the preparation of the manuscript. Without his fundamental work on the measurement of total odd nitrogen and nitrogen oxides in the MOZAIC and IAGOS programmes,
this instrument would not exist. Marlon Tappertzhofen, Torben Blomel, Marcel Berg, Benjamin Winter, Jennifer Gläser and Günther Rupsch are acknowledged for their enormous help to maintain and calibrate the instrument. D. Brunner and H. Ziereis are acknowledged for fruitful discussions and as external reviewers for the SOP. Furthermore, we acknowledge ECMWF for providing meteorological analysis. Part of this project was funded by BMBF in IAGOS-D contract 01LK1301A. The IAGOS database is supported by AERIS (CNES and INSU-
CNRS), where the IAGOS data is stored.

**Competing interests**

The authors declare no competing financial interests.

**Data availability**

The data is available at www.iagos.org.

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



Table 1: Overview of the main instrument components and their specification.

| Part of instrument | Material / Manufacturer | Geometrics V: Volume, L: Length, OD: Outside diameter | Pressure | Residence time in s |
|---|---|---|---|---|
| Inlet tube | FEP | L: 900 mm, OD: 1/8" | ambient | < 0.05 |
| Manifold | Stainless steel | V: 0.3 ml | ambient | 2.5 to 12 |
| Photolytic converter | borosilicate glass | V: 25 ml | ambient | |
| Pre Chamber | Stainless steel | L: 300 to 500 mm, OD: 1/8" | 10 | < 0.04s |
| Reaction chamber | Gold plated stainless steel | V: 28 ml | 10 | |
| Photomultiplier | Hamamatsu R2228P Electron Tubes Enterprises 9828A | - | | |

Table 2: IAGOS $NO_x$ specification

| Quantity | Value |
|---|---|
| Sample flow rate | 150 sccm |
| Inlet flow rate | 1.5 SLM |
| Weight | 29 kg |
| Dimension (LxWxH) | 560x400x283 mm |
| Deployment period | ~ 6 months |
| Time resolution of photon count rate | 10 Hz |



**Table 3: Definition of the different modes of the instruments and their acronyms. Note some of the modes are not used during flight.**

| Air supply | UV-LEDs | Name of the Modes | Comment |
|---|---|---|---|
| **Ambient air** | Off | AA_NO$_{MM}$ | Ambient NO is measured by reaction with $O_3$ in the reaction cell |
| | | AA_NO$_{ZM}$ | About 98% of ambient NO is oxidized in the pre-volume to determine the background signals from other chemical reactions |
| | On | AA_NOc$_{MM}$ | Ambient $NO_x$ (NO + $NO_2$ photolytic reduced) is measured by reaction with $O_3$ in the reaction vessel |
| | | AA_NOc$_{ZM}$ | Same as above flushing the air into the pre-volume |
| **Determine Instrument Background using** | Off | BG_NO$_{MM}$ | Gas bottled synthetic air (Rev.2 instruments) or pure $O_2$ (Rev. 1 instruments), is sucked into the instrument to determine the background signal for ($NO_x$ free gas) of the instrument during flight |
| | | BG_NO$_{ZM}$ | Same as above flushing the air into the pre-volume |
| **Pure $O_2$ or Synthetic air** | On | BG_NOc$_{MM}$ | Gas bottled synthetic air or pure $O_2$ is sucked into the instrument in the reaction vessel to determine the background signal for ($NO_x$ free gas) of the instrument during flight |
| | | BG_NOc$_{ZM}$ | Same as above flushing the air into the pre-volume |
| **Instrumental Stability Check for NO or $NO_2$** | Off | SC_NO$_{MM}$ | Synthetic air (or pure $O_2$) is flushed through a heated (40°C) permutation tube and mixed to the ambient air in the manifold before it is sucked into the instrument directly into the reaction vessel |
| | | SC_NO$_{ZM}$ | Same as above flushing the air into the pre-volume |
| | On | SC_NOc$_{MM}$ | Synthetic air (or pure $O_2$) is flushed through a heated (40°C) permutation tube and mixed to the ambient air in the manifold before it is sucked into the instrument directly into the reaction vessel |
| | | SC_NOc$_{ZM}$ | Same as above flushing the air into the pre-volume |
| **Instrument calibration using external** | Off | Cal_NO$_{MM}$ | Different types of gases (NO, $NO_2$ or $NO_x$ free) can be flushed into the inlet line before it is sucked into the reaction chamber |
| | | Cal_NO$_{ZM}$ | Same as above flushing the air into the pre-volume |
| **gas supplies only in the laboratory** | On | Cal_NOc$_{MM}$ | Different types of gases (NO, $NO_2$ or $NO_x$ free) can be flushed into the inlet line before it is sucked into the reaction chamber |
| | | Cal_NOc$_{ZM}$ | Same as above flushing the air into the pre-volume |

5 **Table 4: Overview of the calibration uncertainties for the two deployment phases in 2015 and 2016.**

| Uncertainty | 2015 | 2016 |
|---|---|---|
| Conversion efficiency | <5% | <4% |
| Detector sensitivity | <2% | <3% |
| Instrument background variability during flight | NO < 10 ppt; $NO_2$ < 20 ppt | |
| Secondary standard | <1% | - |


**Table 5: Criteria for flagging the NOx data accordingly to QA/QC definition in IGAS (www.igas-project.de).**

|  | Value | Comment |
|---|---|---|
| **Good** | **0** | |
| **Limited** | 2 | PMT temperature is larger than -5°C |
|  |  | ozone correction not possible; |
|  |  | water vapor correction not possible; |
|  |  | Large variation of internal stability checks |
| **Erroneous** | 3 | Measurements below the detection limit; |
|  |  | NO night time values enhanced |
|  |  | In-situ zero air measurements are enhanced |
|  |  | PMT temperature is larger than 10°C |
| **Not validated** | 4 | Not validated data points (e. g. $NO_2$ >4 ppb at cruise altitude); |
|  |  | Ascent profile (heating up of the instrument units, e. g. ozone generator); |
| **Missing Value** | 7 | Cyclic measurement of NO and NOx, Zero-Mode, Internal Calibrations |

**Table 6: Life time, mixing ratio and possible interferences of thermal decomposed reservoir species over different temperature ranges. The gray shaded area indicates the most plausible temperature within the $NO_2$ converter in the IAGOS instrument during flight.**

| Species | Life time of the reservoir species at 250 hPa in [s] | | | Interference to $NO_2$ at 250 hPa in [%] | | | Mixing ratio at cruise altitude (source) |
|---|---|---|---|---|---|---|---|
|  | 300 K | 320 K | 340 K | 300 K | 320 K | 340 K | |
| $N_2O_5$ | 23.9 | 2.6 | 0.4 | 11.8 | 68.4 | 100 | < 10 ppt, (Brown et al., 2007) |
| $HO_2NO_2$ | 27.1 | 2.9 | 0.4 | 10.5 | 64.0 | 100 | < 66 ppt (Kim et al., 2007) |
| $CH_3O_2NO_2$ | 1.0 | 0.1 | 0.0 | 94.5 | 100 | 100 | <15 ppt (Browne et al., 2011) |
| $CH_3CO_3NO_2$ =PAN | 1.9E3 | 110 | 9.4 | 0.16 | 2.7 | 27.4 | 300 - 600 ppb (Fischer et al., 2014; Moore and Remedios, 2010) |

**Table 7: Relative amount of available 4 s data points (here NOx) with respect to its validation flag for all flights in 2015 and 2016.**

| Year | Total number | Good (flag=0) | Limited (flag=2) | Error (flag=3) | Invalid (flag=4) |
|---|---|---|---|---|---|
| **2015** | $3.6*10^5$ | 71.7 % | 17.5 % | 3.0 % | 7.8 % |
| **2016** | $4.2*10^5$ | 34.1 % | 58.2 % | 2.9 % | 4.8 % |



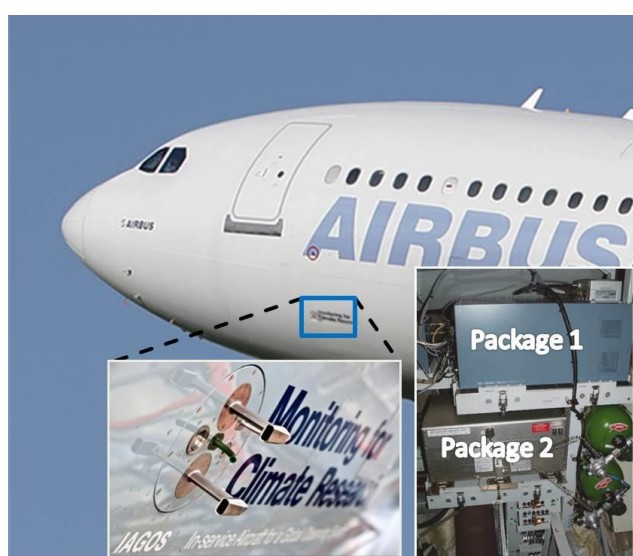

**Figure 1: Position of Package 1 and Package 2 installed aboard the AIRBUS A340-300. The inlet plate including the Rosemount housing is attached at the aircraft skin.**

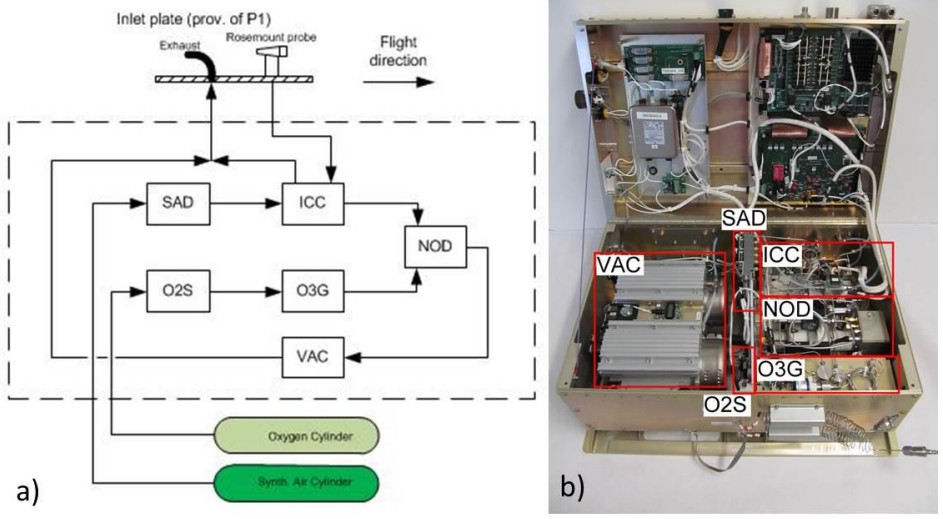

**Figure 2: Schematic diagram of the IAGOS NO$_x$ instrument (Revision 2, certification in progress) showing all connections and modules. A more detailed view is shown in the supplement (Fig. S1). O2S and SAD: Assembly with magnetic valves and capillaries for distribution of oxygen and synthetic air to different parts of the instrument. NOD: Chemiluminescence detector. O3G: ozone generator. VAC: two membrane pumps for the gas flow of the system. ICC:**
10 **Internal calibration and converter unit, containing the manifold, photolytic converter, flow controller and permeation source. In Revision 1 only O$_2$ is used for the internal stability checks during flight, while in Revision 2 this is replaced by synthetic air. O$_2$ is than only used for the ozone generator.**





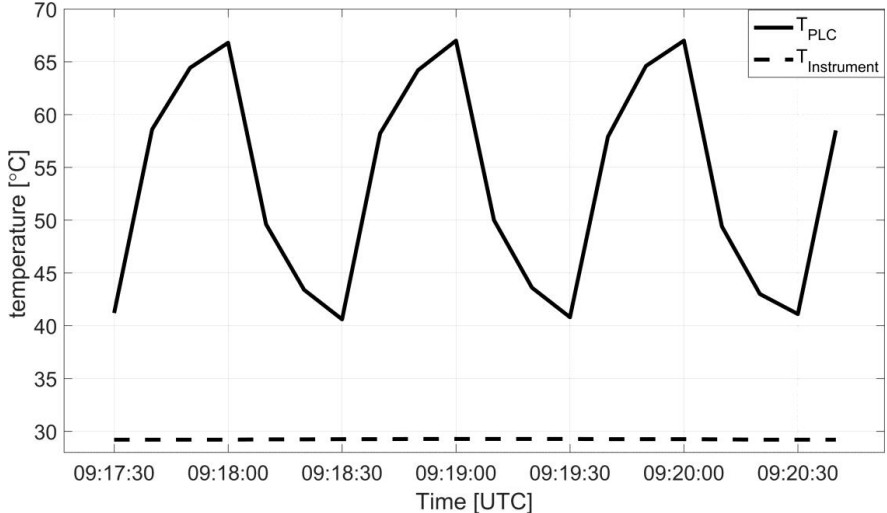

**Figure 3: Gas temperature in the photolytic converter and instrument temperature measured in the laboratory when switching the UV-LEDs on and off every 30s.**

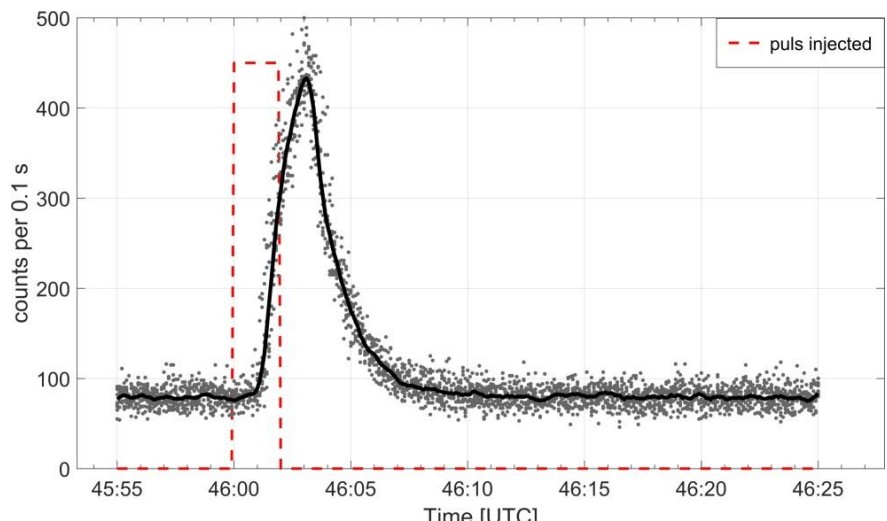

**Figure 4: 10 repetitions of NO pulse (red, dashed) experiment covering 30 s time period. The NO pulse (7.1 ppb) was injected for two seconds directly into the inlet line at each full minute at inlet pressure of 250 hPa. The pulse response (black line) is smooth with a running mean (2s). The width (1/e) of the peak is four seconds and the delay is about 3 seconds.**





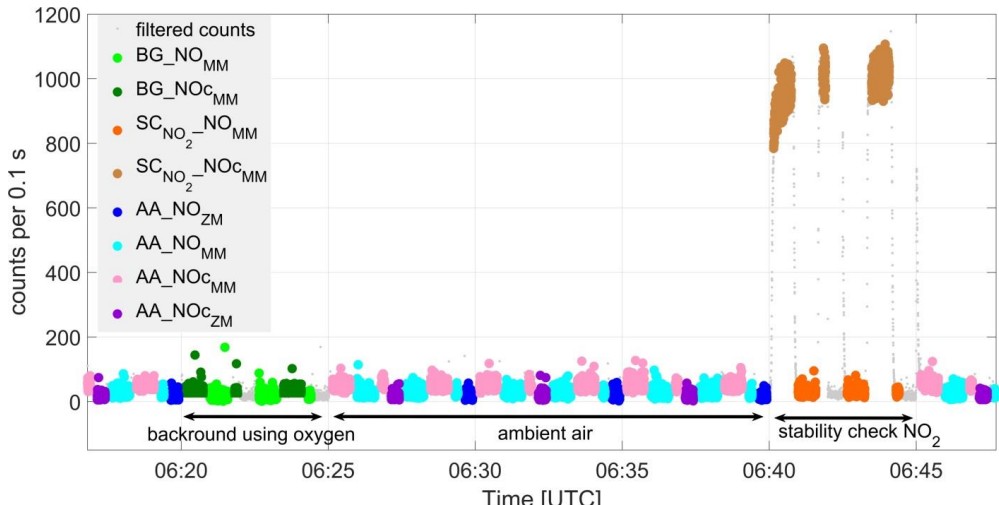

**Figure 5: Example for the in-flight measurement cycle. The different modes of the instrument are denoted by horizontal arrows: During ambient air the measure modes (MM) are shown for NOc (light red) and NO (light blue); the zero modes (ZM) are shown for NOc (purple) and for NO (dark blue). The instrument background checks are made using zero-air gas bottle supply and are shown for NOc (dark green) and for NO (light green); Stability check: using NO₂ produced by the internal calibration source (permeation tube) is shown for NOc (brown) and for NO (orange). The gray dots show discarded data during switching between the different modes.**

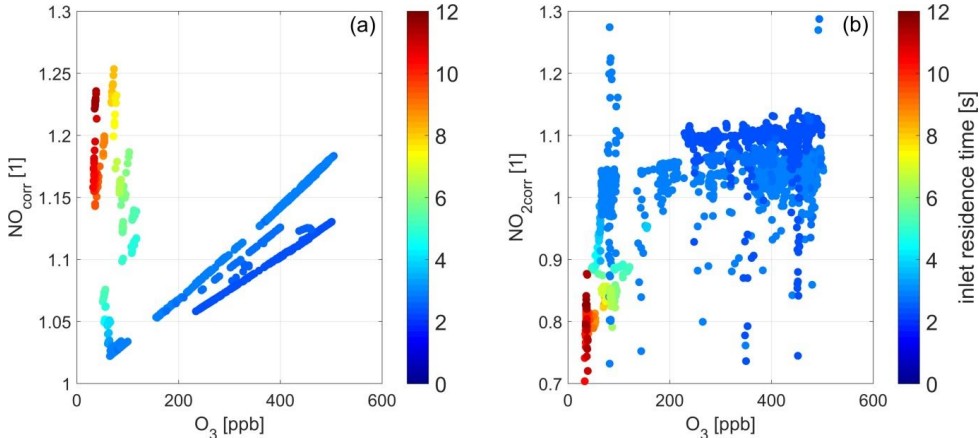

**Figure 6: Typical correction factors for a) NO and b) NO₂, which are depending on ambient ozone and residence time (colorbar) in the inlet-manifold system, for one flight from DUS to NYC in June 2015.**



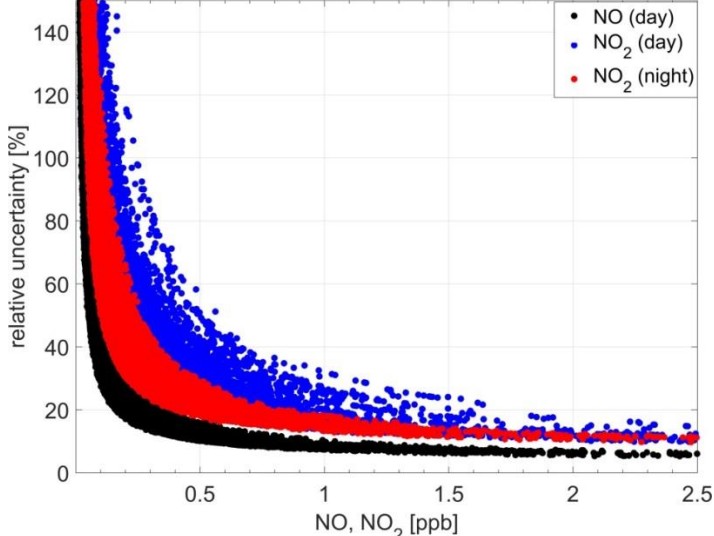

**Figure 7: Relative uncertainty of NO (black, day) and of NO$_2$ (blue, day and red, night) using all measurements (4s) in 2015.**

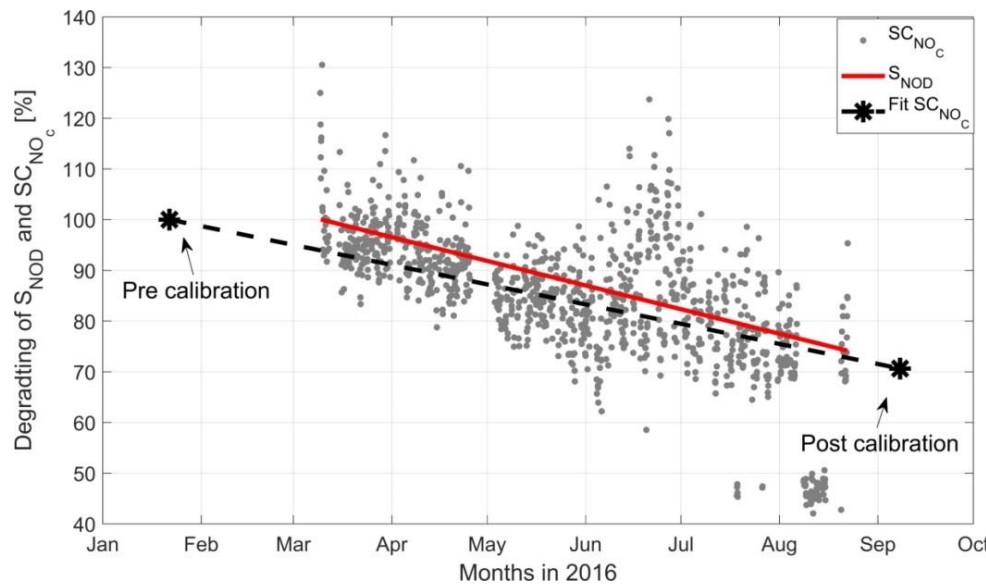

**Figure 8: Linear degrading of the NO-Detector sensitivity (S$_{NOD}$, black) after pre and post calibration in percent. The inflight stability check of NO$_2$ (gray dots) confirms the linear behavior of the degradation of the detector sensitivity during the deployment shown as linear robust fit (red line).**





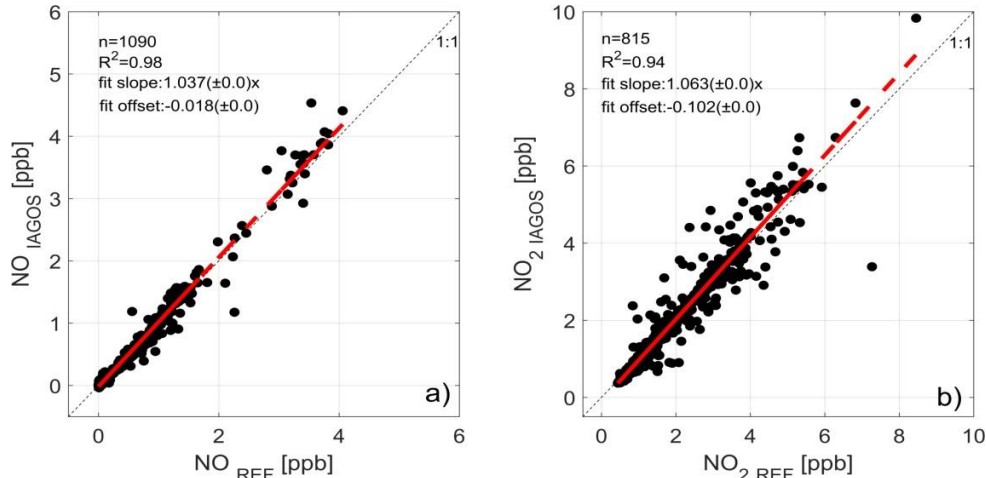

**Figure 9: Two days of ambient NO and NO$_2$ measurements on Mount Hohenpeißenberg in Germany in October 2016 during the ACTRIC s-b-s NOx intercomparison. The data was averaged to 1-min means, no ozone or humidity correction were applied. The reference instrument (REF) was regulary calibrated during the campaign.**

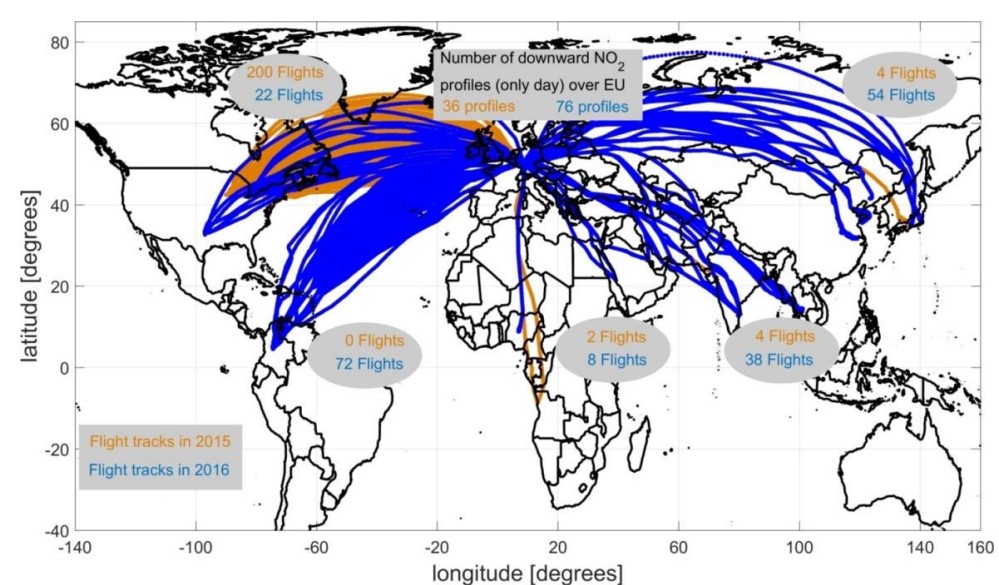

**Figure 10: Flight tracks with the IAGOS NO$_x$ instrument installed aboard the Lufthansa aircraft in 2015 and in 2016. Additionally, the amount of vertical profiles during day is denoted.**


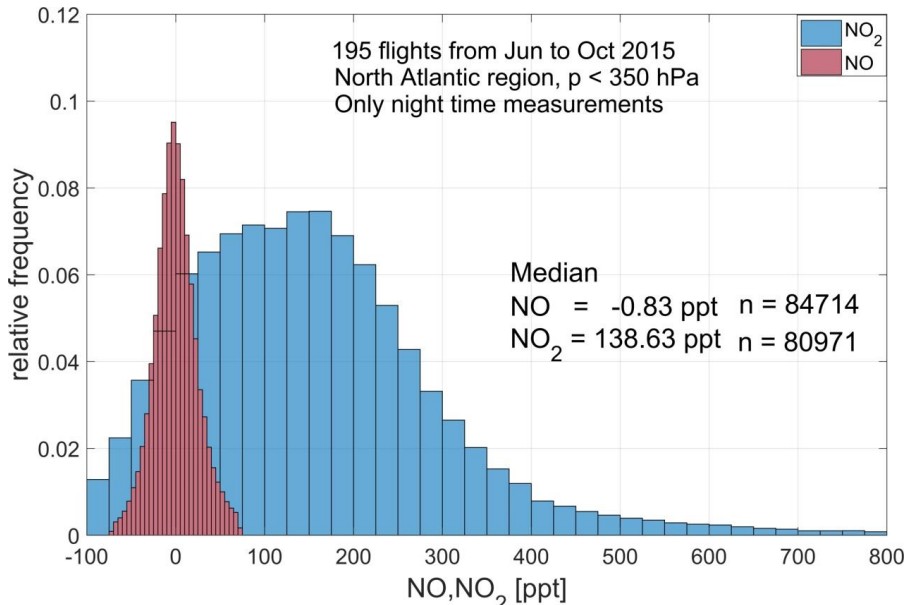

Figure 11: Relative frequency of night time NO and NO₂ measurements (4s) at cruise altitude (p < 350 hPa) from 195 flights over the North Atlantic in 2015. The bin width is 25 ppt for NO₂ and 5 ppt for NO.

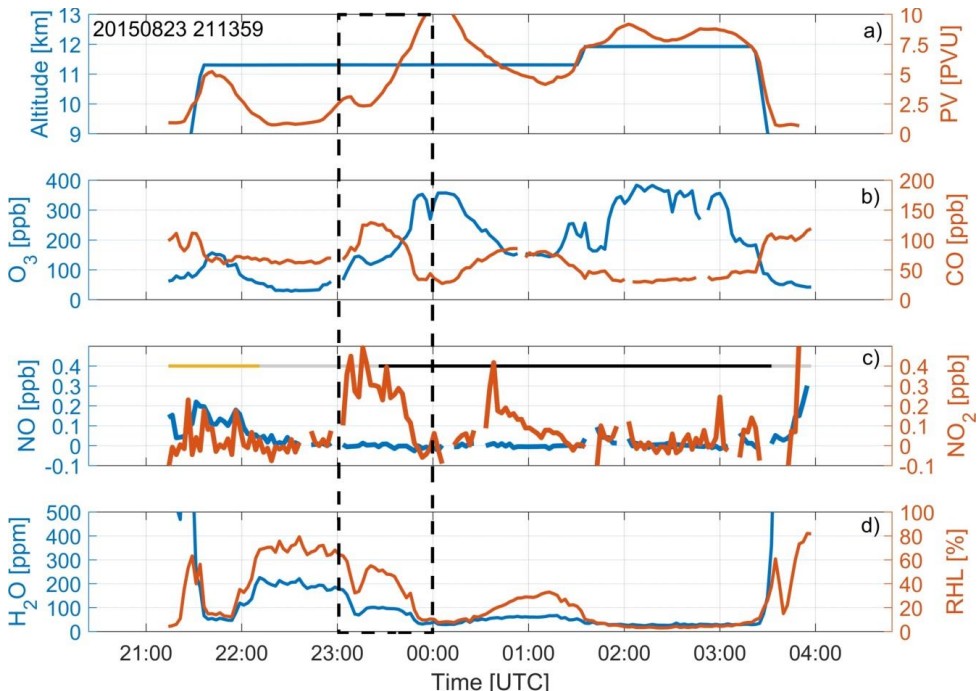

Figure 12: Time series of a) flight altitude and PV, b) ozone and CO, c) NO and NO₂, d) H₂O and RHL from New York City (USA) to Düsseldorf (Germany) on 23. August 2015. The time of day is illustrated as horizontal line (light orange=day, gray, sunset/sunrise, black=night). The shared black box shows a large-scale plume which is discussed in the text. All values are 2-min averages.





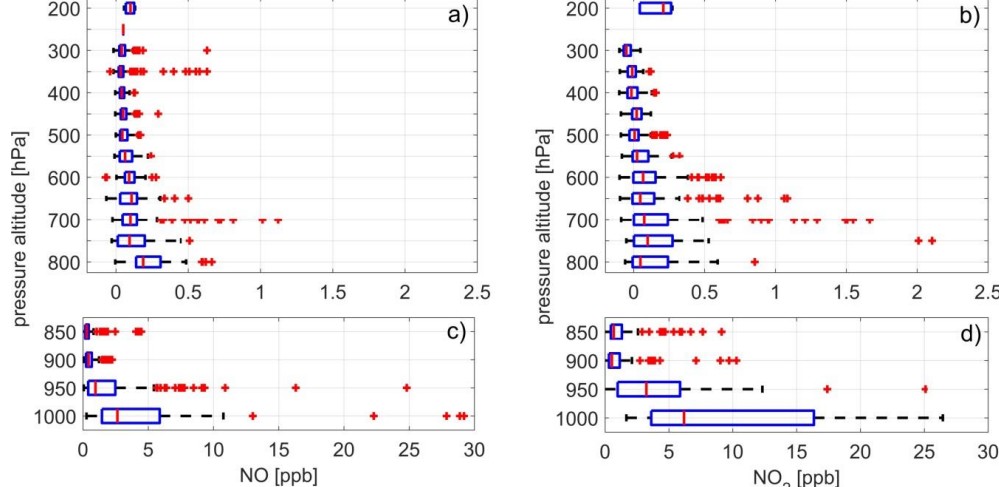

**Figure 13:** Statistical vertical distribution of NO and $NO_2$ (only at day time) of a,c) for NO and b,d) $NO_2$ over Düsseldorf airport in summer (JJA) 2015. Note the different x-axis-scale.