# Peer review of "The IAGOS NOx Instrument – Design, Operation and First Results from Deployment aboard Passenger Aircraft"

_Atmospheric Measurement Techniques, 2017_

## Referee Comment (RC1) · Anonymous Referee #1 · 15 Feb 2018

In "The IAGOS $NO_X$ Instrument – Design, Operatio and First Results from Deployment aboard Passenger Aircraft", Berkes et al. describe the $NO_X$ instrument deployed on a Lufthansa aircraft and provide details about the measurement technique and limitations. The authors carefully step through the various calibration and zeroing techniques, as well as data reduction steps. The details provided will be extremely helpful to scientists looking at any IAGOS generated $NO_X$ data in the future.

Overall, the paper is scientifically solid. It would benefit from some rewriting for clarity and addressing the following issues:

Major points:

Calibrations and corrections for vertical profiles: All calibrations are done at 250 hPa inlet pressure, but data is presented from vertical profiles. No analysis was presented justifying whether this calibration should hold at the higher pressures during the landing approach. The same holds for the instrument response characterization. The authors should either show why neither of these factors change with altitude, or should account for those changes in the error analysis and present a pressure dependent error.

Total uncertainty: The total uncertainty does not account for uncertainty in some of the corrections applied to the data. Perhaps most importantly, the authors state on p11, L1 "the ozone correction is very sensitive to the ozone mixing ratio", but don't account for ozone mixing ratio in the total error analysis. If the correction is that sensitive, there needs to be more discussion about the ozone measurement. What is the error of that measurement? Also, are the instruments perfectly synchronized in time or could there be a small offset altering that correction? This is also an issue with the instrument drift during deployment. The authors show the drift is linear, but show two different linear fits. It is not clear which of those linear fits is actually used during analysis and how much it might matter if the other fit was used.

There are some writing and organization issues that make the paper difficult to read. I have noted several in the "minor points" below, but a more thorough editing would be beneficial.

Minor points:

P2,L8. "whereas" does not make sense here
P2,L12. The list is presented unclearly and the sentence should be rewritten
P2,L26. "Despite the progress..." should start a new paragraph
P2,L33-38. This paragraph is unclear.
P5,L36. It is unclear what the sentence beginning with "However" is actually about (e.g., conversion efficiency?).
P6,L4. Change to "90 cm long PFA tube with a diameter..."
P6,L10. "trough" should be "through"
P6,L18. $O_3$ needs subscript
P8,L5. Change with to within

P10. Move these correction to before the steps on P9 that use them.

P11,L15-16. It is not clear what the second LOD numbers, presented in parentheses, are.

Section 4.3. Throughout this section there are percents presented, but it is not clear whether those are percent of the measured NO or percent of the interfering species (e.g., HONO).

P13,L3. The concentrations are "too small" for what?

P14,L3. It reads that NO has a "variation" of 25 ppt, but it looks much larger on the graph. Do you mean a standard deviation?

P14,L37-39. It is not clear which $NO_2$ peak is referred to here. Is this what is in the black box on the figure? Most of this paragraph is confusing.

P15,L2. $O_3$ reaches over 350 ppbv in the figure presented. Not just 200 ppbv.

P15,L10. Correct plum to plume.

P15,L32. Correct averaged to average

P15,L33. $NO_2$ is in parentheses after NO, but no $NO_2$ data are presented.

---

## Referee Comment (RC2) · Anonymous Referee #2 · 23 Feb 2018

The submission by Berkes et al. presents a thorough description of a NOx instrument that has flown routinely on commercial aircraft as part of a larger package with several other instruments. The measurement is based on the chemiluminescent detection of NO and NO2 (after photolysis to NO). The photolytic converter for NO2 is a major improvement over prior instrumentation used in such flights. The instrument is thoroughly characterized, and some representative measurement results are presented. The paper can be published essentially as is, though some minor points should be addressed. Also, although perfectly clear, the English could be improved in spots. A few representative examples are noted below, but by no means complete.

[Figure]

Minor points:

p.2, lines 4-5: please explain why production rate most favorable in the UT. Is this in regard to efficiency or total amount produced? Why not more favorable where heavily polluted? Rate is higher there.

p.4, line 22: NOD not yet defined.

p.5 line 25: Is the 18 kV AC or DC. If AC, what is frequency?

p.7, line 32: Would be useful to cite numerical value for sensitivity.

p.10, line 33: Better to say O3 concentration (in cmˆ-3) rather than mixing ratio (dimensionless).

p.11, line 27: depending / change to dependent

p.11, line 30: An uncertainty in NO2 not acknowledged is that due to the use an NO value that is not simultaneous with NOx detection. NO2 error can be much larger if mixing ratios are varying, when NO is uncertain.

p.14, line 26: agree / change to agrees

P.14 line 28: "with to" / change to "by", "by" / change to "from"

p.15, line 10: typo: "plum"

Fig. 12, the right side of the box for the plume could probably be shifted left about 15 minutes.

p.16, line 5: depending / change to dependent

p.16, Line 13: units of sensitivity?

---

## Author Comment (AC1) · 18 Apr 2018

**Reply to**

In "The IAGOS NOX Instrument – Design, Operation and First Results from Deployment aboard Passenger Aircraft", Berkes et al. describe the NOX instrument deployed on a Lufthansa aircraft and provide details about the measurement technique and limitations. The authors carefully step through the various calibration and zeroing techniques, as well as data reduction steps.
The details provided will be extremely helpful to scientists looking at any IAGOS generated NOX data in the future.

Overall, the paper is scientifically solid. It would benefit from some rewriting for clarity and addressing the following issues:

**We thank the referee for her/his comments, which we address (in bold) point by point in our reply below.**

Major points:

Calibrations and corrections for vertical profiles: All calibrations are done at 250 hPa inlet pressure, but data is presented from vertical profiles. No analysis was presented justifying whether this calibration should hold at the higher pressures during the landing approach. The same holds for the instrument response characterization. The authors should either show why neither of these factors change with altitude, or should account for those changes in the error analysis and present a pressure dependent error.

**The reviewer is correct that the different flight altitudes need to be considered. However, as we stated the in section 2.2.2, we don't use a constant conversion efficiency to calculate the mixing ratio of NOx and NO2. For each data point the conversion efficiency is calculated depending on the ambient pressure (see figure below). This is also the case for the ozone correction etc. To clarify the issue we added the following sentence to the manuscript (line 17-21 at page 12 of the annotated manuscript:**

**"Since the ozone correction is sensitive to the ozone mixing ratio, the residence time $\tau$ inside the PLC is determined for each instrument for the expected pressure range from 1000 hPa to 180 hPa, which provides the correction function $\tau(p)$ to be used in Eqs. 10 and 11 (see Fig. S5 in the supplement material). For the future generation of IAGOS NOx instruments, we plan to keep the residence time in the PLC at 3 s, independent from the inlet pressure, by using a critical nozzle."**

[Figure]

Total uncertainty: The total uncertainty does not account for uncertainty in some of the corrections applied to the data. Perhaps most importantly, the authors state on p11, L1 "the ozone correction is very sensitive to the ozone mixing ratio", but don't account for ozone mixing ratio in the total error analysis. If the correction is that sensitive, there needs to be more discussion about the ozone measurement. What is the error of that measurement? Also, are the instruments perfectly synchronized in time or could there be a small offset altering that correction? This is also an issue with the instrument drift during deployment. The authors show the drift is linear, but show two different linear fits. It is not clear which of those linear fits is actually used during analysis and how much it might matter if the other fit was used.

**We apologize to the reviewer that our statement leads to a misunderstanding and improved the manuscript at several places.**
**The residence time of the sampled air mass within the converter plays the most important role for the NO correction (see Fig. 7). The uncertainty of the ozone measurements is given with 2 ppbv ± 2% (Nédélec et al., 2015).**
**Considering this uncertainty, the ozone correction factor would change within the planetary boundary layer (900 hPa, 293K) from 1.16 to 1.17 at ozone mixing ratios of 40 or 43 ppbv, respectively. The impact of the ozone uncertainty is therefore only 1%. Therefore we believe that including the ozone uncertainty in the total uncertainty for NO is negligible. As we wrote in the manuscript, we plan to keep the residence time of the sampled air mass at 3 s for all ambient pressure conditions in a future revision of the instrument. With this, the ozone correction factor (900 hPa, 293K, 40 ppb of ozone) will reduce from 1.16 to 1.05.**

**The instrument is synchronized during flight with the main package P1. The time synchronization has been cross-checked using the ozone measurements from P1, which are also transferred every 4s to the P2 instrument.**

**The drift of the detector sensitivity is determined using the pre and post calibrations in the laboratory. The additional fit in figure 8 from the internal quality checks is just shown to justify our procedure.**

There are some writing and organization issues that make the paper difficult to read. I have noted several in the "minor points" below, but a more thorough editing would be beneficial.

Minor points:
P2,L8. "whereas" does not make sense here
**Changed to "whereas" to "also"**

P2,L12. The list is presented unclearly and the sentence should be rewritten.
**We rewrote the sentence.**

P2,L26. "Despite the progress…" should start a new paragraph
**Done**

P2,L33-38. This paragraph is unclear.
**We moved the paragraph to another position. With it we wanted to provide some information about the current state of aircraft and satellite missions and the performance of model simulations.**

P5,L36. It is unclear what the sentence beginning with "However" is actually about (e.g., conversion efficiency?).
**We rephrased this paragraph.**

P6,L4. Change to "90 cm long PFA tube with a diameter…"
**Done**

P6,L10. "trough" should be "through"
**Done**

P6,L18. O3 needs subscript
**Done**

P8,L5. Change with to within
**Done**

P10. Move these correction to before the steps on P9 that use them.
**We disagree with the reviewer and think that the water - and ozone corrections needs to be explained carefully and therefore we placed them just after the outline of the data processing.**

P11,L15-16. It is not clear what the second LOD numbers, presented in parentheses, are.
**The numbers in the parentheses shall indicate that the detection limit is depending on the sensitivity of the detector. We removed them and included an additional sentence to be clearer.**

Section 4.3. Throughout this section there are percents presented, but it is not clear whether those are percent of the measured NO or percent of the interfering species (e.g., HONO).
**We improved this section.**

P13,L3. The concentrations are "too small" for what?
**Changed to "too low" to have a major impact to the NO2 measurements.**

P14,L3. It reads that NO has a "variation" of 25 ppt, but it looks much larger on the graph. Do you mean a standard deviation?
**Yes we meant the standard deviation. We corrected that sentence.**

P14,L37-39. It is not clear which NO2 peak is referred to here. Is this what is in the black box on the figure? Most of this paragraph is confusing.
**We rephrase this paragraph and apologies for the confusing.**

P15,L2. O3 reaches over 350 ppbv in the figure presented. Not just 200 ppbv.
**We rephrase this paragraph and apologies for the confusing.**

P15,L10. Correct plum to plume.
**Done**

P15,L32. Correct averaged to average
**Done**

P15,L33. NO2 is in parentheses after NO, but no NO2 data are presented.
**We deleted the parentheses.**

---

## Author Comment (AC2) · 18 Apr 2018

The submission by Berkes et al. presents a thorough description of a NOx instrument that has flown routinely on commercial aircraft as part of a larger package with several other instruments. The measurement is based on the chemiluminescent detection of NO and NO2 (after photolysis to NO). The photolytic converter for NO2 is a major improvement over prior instrumentation used in such flights. The instrument is thoroughly characterized, and some representative measurement results are presented.

The paper can be published essentially as is, though some minor points should be addressed. Also, although perfectly clear, the English could be improved in spots. A few representative examples are noted below, but by no means complete.

**We thank the referee for her/his comments, which we address (in bold) point by point in our reply below.**

Minor points:
p.2, lines 4-5: please explain why production rate most favorable in the UT. Is this in regard to efficiency or total amount produced? Why not more favorable where heavily polluted? Rate is higher there.
**We have removed this sentence because it adds confusion and is not needed for our arguments.**

p.4, line 22: NOD not yet defined.
**Corrected**

p.5 line 25: Is the 18 kV AC or DC. If AC, what is frequency?
**The high-voltage transformer is operated by a pulsing DC source, running at 250 Hz. The HV transformer thus generates 18 kV at a frequency of 250 Hz. We added the frequency value to the text.**

p.7, line 32: Would be useful to cite numerical value for sensitivity.
**We included:**
**"As an example, for a detector sensitivity of 1000 cps pptv$^{-1}$ the uncertainty is 30 cps pptv$^{-1}$. Please note, the detector sensitivity is not a constant value and it decreases during the deployment."**

p.10, line 33: Better to say O3 concentration (in cm^-3) rather than mixing ratio (dimensionless).
**This is correct. We convert the mixing ratio in concentration. We changed this in the text.**

p.11, line 27: depending / change to dependent
**Changed**

p.11, line 30: An uncertainty in NO2 not acknowledged is that due to the use an NO value that is not simultaneous with NOx detection. NO2 error can be much larger if mixing ratios are varying, when NO is uncertain.

**The reviewer is right that we cannot provide simultaneously NO and NOx measurements with the IAGOS NOx instrument. However, during night time NO is converted to NO2 which is therefore measured via NOx.**

**In Fig. 7 we show the uncertainty for NO2 for day and night time, where we tried to demonstrate that the uncertainty for NO2 is larger during day time, when NO is not zero.**

**At the current stage, the instrument switches between the NO and NOx mode every 30s. Each NO2 data point is calculated by subtracting the median of the NO measurements before and after each NOx cycle. We cannot provide a better estimate for the "true" NO value and this makes it even more difficult to estimate an uncertainty for NO2.**

p.14, line 26: agree / change to agrees
**Changed**

P.14 line 28: "with to" / change to "by", "by" / change to "from"
**Changed**

p.15, line 10: typo: "plum"
**Corrected**

Fig. 12, the right side of the box for the plume could probably be shifted left about 15 minutes.
**We changed the range of box.**

p.16, line 5: depending / change to dependent
**Changed**

p.16, Line 13: units of sensitivity?
**It is counts per second per pptv (cps pptv$^{-1}$)**